# Tree-ring isotopes capture interannual vegetation productivity dynamics at the biome scale

Mathieu Levesque[1,2], Laia Andreu-Hayles[2], William Kolby Smith [3], A. Park Williams[2], Martina L. Hobi[4,5], Brady W. Allred[6,7] & Neil Pederson [8]

Historical and future trends in net primary productivity (NPP) and its sensitivity to global change are largely unknown because of the lack of long-term, high-resolution data. Here we test whether annually resolved tree-ring stable carbon ($\delta^{13}C$) and oxygen ($\delta^{18}O$) isotopes can be used as proxies for reconstructing past NPP. Stable isotope chronologies from four sites within three distinct hydroclimatic environments in the eastern United States (US) were compared in time and space against satellite-derived NPP products, including the long-term Global Inventory Modeling and Mapping Studies (GIMMS3g) NPP (1982–2011), the newest high-resolution Landsat NPP (1986–2015), and the Moderate Resolution Imaging Spectro-radiometer (MODIS, 2001–2015) NPP. We show that tree-ring isotopes, in particular $\delta^{18}O$, correlate strongly with satellite NPP estimates at both local and large geographical scales in the eastern US. These findings represent an important breakthrough for estimating inter-annual variability and long-term changes in terrestrial productivity at the biome scale.

[1] Forest Management Group, Department of Environmental Systems Science, Institute of Terrestrial Ecosystems, ETH Zurich, 8092 Zurich, Switzerland. [2] Tree-Ring Laboratory, Lamont-Doherty Earth Observatory of Columbia University, Palisades, NY 10964, USA. [3] School of Natural Resources and the Environment, University of Arizona, Tucson, AZ 85721, USA. [4] SILVIS Lab, Department of Forest and Wildlife Ecology, University of Wisconsin-Madison, Madison, WI 53706, USA. [5] WSL Swiss Federal Institute for Forest, Snow and Landscape Research, 8903 Birmensdorf, Switzerland. [6] W.A. Franke College of Forestry and Conservation, University of Montana, Missoula, MT 59812, USA. [7] Numerical Terradynamic Simulation Group, University of Montana, Missoula, MT 59812, USA. [8] Harvard Forest, Harvard University, Petersham, MA 01366, USA. Correspondence and requests for materials should be addressed to M.L. (email: mathieu.levesque@usys.ethz.ch) or to L.A.-H. (email: lah@ldeo.columbia.edu)

Recent achievements in satellite and model estimates of NPP have improved our understanding of the influence of climate change and increased atmospheric $CO_2$ concentration ($_{atm}CO_2$) on terrestrial NPP[1,2]. However, decadal to centennial trends in historical NPP have been difficult to characterize because of the lack of high-resolution and long-term data. NPP estimates from satellite data only cover the last three decades and long-term records rely on integration of data from multiple instruments, which introduces substantial uncertainty in understanding trends[3]. The longest flux-tower measurements cover only the last 25 years[4] and estimates of NPP productivity from forest inventories can cover longer time periods, but often have multiannual to decadal gaps between consecutive measurements.

Perhaps even more relevant to NPP trends and dynamics, there is an incomplete understanding of how the factors that influence NPP[5] interact with one another and whether these interactions are synergistic or antagonistic at multi-annual to decadal scales. Among current anthropogenic effects, increased growing season length, moderate warming, and elevated $_{atm}CO_2$ may enhance NPP by stimulating photosynthetic assimilation rates, particularly in locations where precipitation increases[6]. Yet, drought[2,7] and limited nutrient availability[8] may reduce NPP and carbon storage. Therefore, long-term patterns in NPP, which integrate complex interactions between biophysical (water and temperature) and biogeochemical ($CO_2$ and soil nutrients) factors, are difficult to capture with satellite observations and current Earth system models[9,10]. It is crucial that we find new proxies that can provide retrospective estimates of vegetation productivity at high temporal resolution across a range of spatial scales.

Tree-ring width and $\delta^{13}C$ and $\delta^{18}O$ signatures recorded in tree rings can provide long-term retrospective information at subannual and annual resolution on physiological and environmental processes and interactions between tree productivity and water use[11–13]. Tree-ring width chronologies have long been used for reconstructing past climate[14,15], forest dynamics[16], climatic sensitivity[17], and more recently, for assessing carbon sequestration and forest productivity[18–21]. Still, current protocols require the sampling of hundreds of trees over tens of sites to yield regional-scale estimates of productivity[18,22]. A recent study indicates that tree-ring $\delta^{13}C$ could serve as a measure of forest productivity locally[23]. However, the potential of tree-ring isotopes for inferring regional to continental terrestrial vegetation productivity has never been tested.

In contrast to the other NPP proxies, tree-ring isotopic ratios vary depending on photosynthesis and stomatal conductance rates at the leaf level[24,25], physiological processes that influence tree carbon gain and loss, and ultimately, vegetation productivity. Photosynthesis and stomatal conductance are regulated by physiological processes and strongly influenced by climatic conditions such as irradiance, relative humidity, temperature, and soil moisture[25,26]. Therefore, since tree-ring isotopic ratios and vegetation productivity are intrinsically associated, a strong agreement between isotopic composition in tree rings and NPP is expected.

To test whether tree-ring isotopes can be used as proxies for NPP at large spatial scales, we compared newly-developed high-resolution tree-ring carbon and oxygen isotopic records from four regions in the eastern US (Fig. 1; Supplementary Table 1) with the latest generations of satellite-derived NPP estimates. We used the GIMMS3g NPP dataset (1982–2011)[1], the newest high-resolution Landsat (1986–2015), and the MODIS (2001–2015) NPP products available for the conterminous United States[27]. We sampled trees of two abundant and representative species of the eastern US deciduous broadleaf forest, *Liriodendron tulipifera* L. and *Quercus rubra* L.[28], at sites located at the opposite ends of the species' distributions (Supplementary Fig. 1) and with contrasting

hydroclimatic trends (Fig. 1). We sampled a site in the northeastern US, a region exhibiting strong wetting and mild warming trends and three sites located in the southeast and central US where warming has not yet taken place[29], but where summer precipitation trends are spatially variable. Our sampling scheme allowed us to test whether tree-ring proxies can be used to infer NPP under wide-ranging environmental conditions.

We find that tree-ring carbon and oxygen isotopes record satellite NPP estimates at local and regional scales. The results are generally consistent among the studied tree species and under contrasting hydroclimatic conditions across the eastern US deciduous broadleaf forest. This new approach aims to provide a step forward in estimating historical changes in NPP at the biome scale, which may then prove critical for parameterization of global terrestrial carbon ecosystem models.

## Results

**NPP signals recorded in tree-ring width and stable isotopes.** The $\delta^{18}O$ tree-ring chronologies correlated significantly with the three NPP datasets at local (i.e., the nearest grid cell to each study site) and regional scale (i.e., field correlations) in the eastern US (Fig. 2a). At local scale, we found significant negative correlations ($\rho = -0.49$ to $-0.56$; $P \leq 0.006$) between the $\delta^{18}O$ data and GIMMS3g NPP time series (1982–2011). Significant negative correlations ($\rho = -0.41$ to $-0.71$; $P \leq 0.025$) were also locally found between site-level $\delta^{18}O$ chronologies and Landsat NPP time series (1986–2015). When the same analyses were conducted with the MODIS NPP for the period 2001–2015, significant correlations were found for two of the sites, Frick Creek and Ouachita Forest ($\rho = -0.65$ and $-0.67$, respectively; $P \leq 0.004$). Chronologies of $\delta^{18}O$ shared very similar interannual variability and trends with GIMMS3g and Landsat NPP time series (Fig. 3a,b). Particularly, $\delta^{18}O$ at Black Rock Forest tracked very well the positive shift in NPP that has occurred after 1990. Overall, the field correlation maps indicated a strong agreement between tree-ring $\delta^{18}O$ and satellite NPP gridded estimates across large geographical regions surrounding the study sites (Fig. 2a).

The $\Delta^{13}C$ tree-ring chronologies also correlated significantly with the satellite NPP products. GIMMS3g NPP correlated significantly with $\Delta^{13}C$ at Crowley's Ridge at the local ($\rho = 0.44$; $P = 0.016$) and regional scale in central US (Fig. 2b). Regarding the Landsat NPP data, very strong correlations were found locally with the $\Delta^{13}C$ chronology at Black Rock Forest ($\rho = 0.72$; $P < 0.001$, Fig. 2b). Although lower correlation values were found between the $\Delta^{13}C$ chronologies at Black Rock Forest ($\rho = 0.47$; $P = 0.090$) and Ouachita Forest ($\rho = 0.52$; $P = 0.048$) and the MODIS NPP data, the field correlation maps indicated a strong agreement between $\Delta^{13}C$ and NPP at the regional scale (Fig. 2b).

Overall, a strong agreement in interannual and decadal variability was found between the isotopic tree-ring chronologies ($\delta^{18}O$ and $\Delta^{13}C$) and the three NPP products (GIMMS3g, Landsat and MODIS) at local and regional scales. These results are consistent for the two studied species *Liriodendron tulipifera* and *Quercus rubra* as demonstrated by the spatial correlation maps done individually at Black Rock Forest and Frick Creek where the two species were present (Supplementary Fig. 2).

In contrast to the isotopic records, the tree-ring width chronologies showed very weak associations with the satellite NPP datasets (Fig. 2c). No significant correlations were observed between Frick Creek, Crowley's Ridge and Ouachita Forest tree-ring width chronologies and any of the three NPP products. The exception was the Black Rock Forest tree-ring width chronology that showed significant positive correlations at local scale with Landsat and GIMMS NPP ($\rho = 0.59$ and 0.38, respectively; $P \leq$

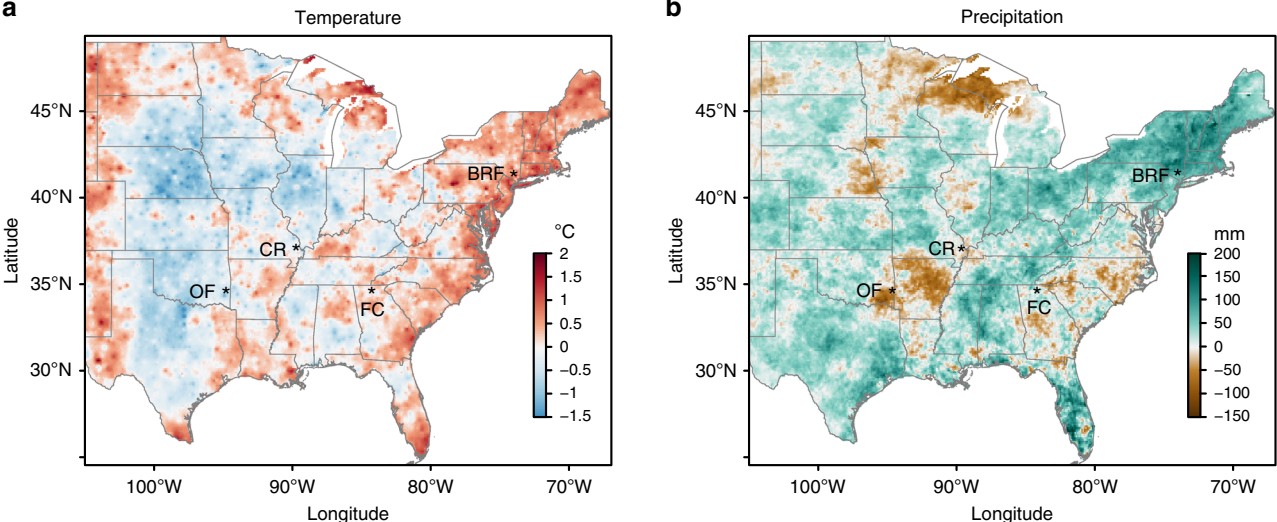

**Fig. 1** Location of the study sites and climatic trends in the eastern United States for 1950–2015. Linear change in summer (June, July and August) temperature (**a**) and precipitation (**b**). Black Rock Forest NY (BRF); Frick Creek GA (FC); Crowley's Ridge MO (CR); and Ouachita Forest OK (OF). Gridded 4-km resolution summer temperature and precipitation data were obtained from the PRISM Climate Group, Oregon State University (http://prism.oregonstate.edu)

0.041) and at regional level with both Landsat and MODIS NPP data (Fig. 2c).

**Climate signals and canopy processes recorded in tree rings**. For 1982–2015, $\delta^{18}O$ and $\Delta^{13}C$ chronologies correlated significantly and strongly to growing season vapor pressure deficit (VPD) at local and regional scale (Fig. 4a, b). Consistently, when we extended the analysis to the period 1950–2015 (Fig. 3c), the correlations between $\delta^{18}O$ and VPD were also very strong at the four study sites. In particular, at Black Rock Forest ($\rho = 0.69$, $P \leq 0.001$) and Frick Creek ($\rho = 0.75$, $P \leq 0.001$) where the $\delta^{18}O$ and VPD time series co-varied closely at interannual and interdecadal timescales. In accordance, tree-ring $\delta^{18}O$ and $\Delta^{13}C$ correlated strongly with growing season precipitation at Frick Creek and Black Rock Forest. However, correlations between tree-ring isotopes and precipitation were weaker than with VPD at the regional scale (Fig. 4a, b). Tree-ring width chronologies showed only significant correlations with VPD at Black Rock Forest and Ouachita Forest (Fig. 4c). No correlations were found between the tree-ring variables and the growing season minimum temperature across all sites (Fig. 4).

In contrast to the strong agreement found between tree-ring isotopic data and moisture availability expressed by VPD (Fig. 3), none of the tree-ring chronologies (tree-ring width, $\Delta^{13}C$, and $\delta^{18}O$) exhibited any significant correlation with the absorbed fraction of photosynthetically active radiation (FPAR3g) and leaf area index (LAI3g) used to produce the GIMMS3g NPP product (Supplementary Fig. 3). While FPAR3g and LAI3g datasets had no agreement with tree-ring data for the period 1982–2011, the MODIS FPAR and LAI showed slightly different results. The $\delta^{18}O$ and $\Delta^{13}C$ chronologies at Black Rock Forest correlated significantly to MODIS FPAR and LAI at the local scale ($\delta^{18}O$: $\rho = 0.52$ to $0.57$, $P < 0.05$; $\Delta^{13}C$: $\rho = -0.68$ to $-0.73$, $P < 0.01$) and regionally across most of the northeastern US (Supplementary Fig. 4). Some significant correlations ($\rho = 0.58$ to $0.72$, $P < 0.05$) were also found between the $\delta^{18}O$ time series at Frick Creek and Ouachita Forest and MODIS FPAR and LAI data.

**Discussion**

Supporting our hypothesis, we found significant correlations between annually resolved $\Delta^{13}C$ and $\delta^{18}O$ tree-ring data and annual NPP data across large geographical regions with differing hydroclimatic conditions. Despite distinct environmental conditions and diverging trends in climate, these results indicate a strong agreement between tree-ring $\delta^{18}O$ and space-based NPP estimates across the US broadleaf forest biome (Fig. 2a). While the northeastern US region exhibits strong wetting and mild warming trends, the southeast/central US region is known as a "warming hole" where the increase in global temperature has not yet taken place[29], but where summer precipitation trends have been geographically variable (Fig. 1). Irrespective of site moisture conditions and climatic trends, $\delta^{18}O$ tree-ring data strongly correlated to GIMMS3g NPP (1982–2011), Landsat NPP (1986–2015), and MODIS NPP (2001–2015) datasets (Fig. 2a). This persistent coupling between $\delta^{18}O$ tree-ring series and satellite NPP data in the context of varying climate (trends and inter-annual/decadal variability) is partly explained by the strong association between $\delta^{18}O$ and VPD (Figs 3c, 4a).

Daily VPD, together with minimum temperature, are the climate components of the satellite productivity products[26], which also include other factors such as the absorbed fraction of photosynthetically active radiation (FPAR) and the leaf area index (LAI), as well as solar radiation and land cover[27] (see Methods). While $\delta^{18}O$ was strongly related to both VPD and NPP, very weak associations were found with minimum temperatures (Fig. 4a). It is well established that the $\delta^{18}O$ ratio in cellulose is sensitive to relative humidity and air temperature, which when their individual effects are integrated, correspond to the leaf-to-air vapor pressure difference (i.e., VPD)[30]. The strong control that VPD exerts on gas exchange at the leaf level and subsequently on growth of individual trees ultimately translates to changes in productivity at the ecosystem level[11,30]. Regarding $\Delta^{13}C$ tree-ring data, while strong and significant correlations were found with Landsat and MODIS NPP (Fig. 2b) and VPD (Fig. 4b) locally and regionally at Black Rock Forest, correlations over the other sites were not as strong as those for $\delta^{18}O$. Overall, the strong correspondence between tree-ring $\delta^{18}O$ data and NPP

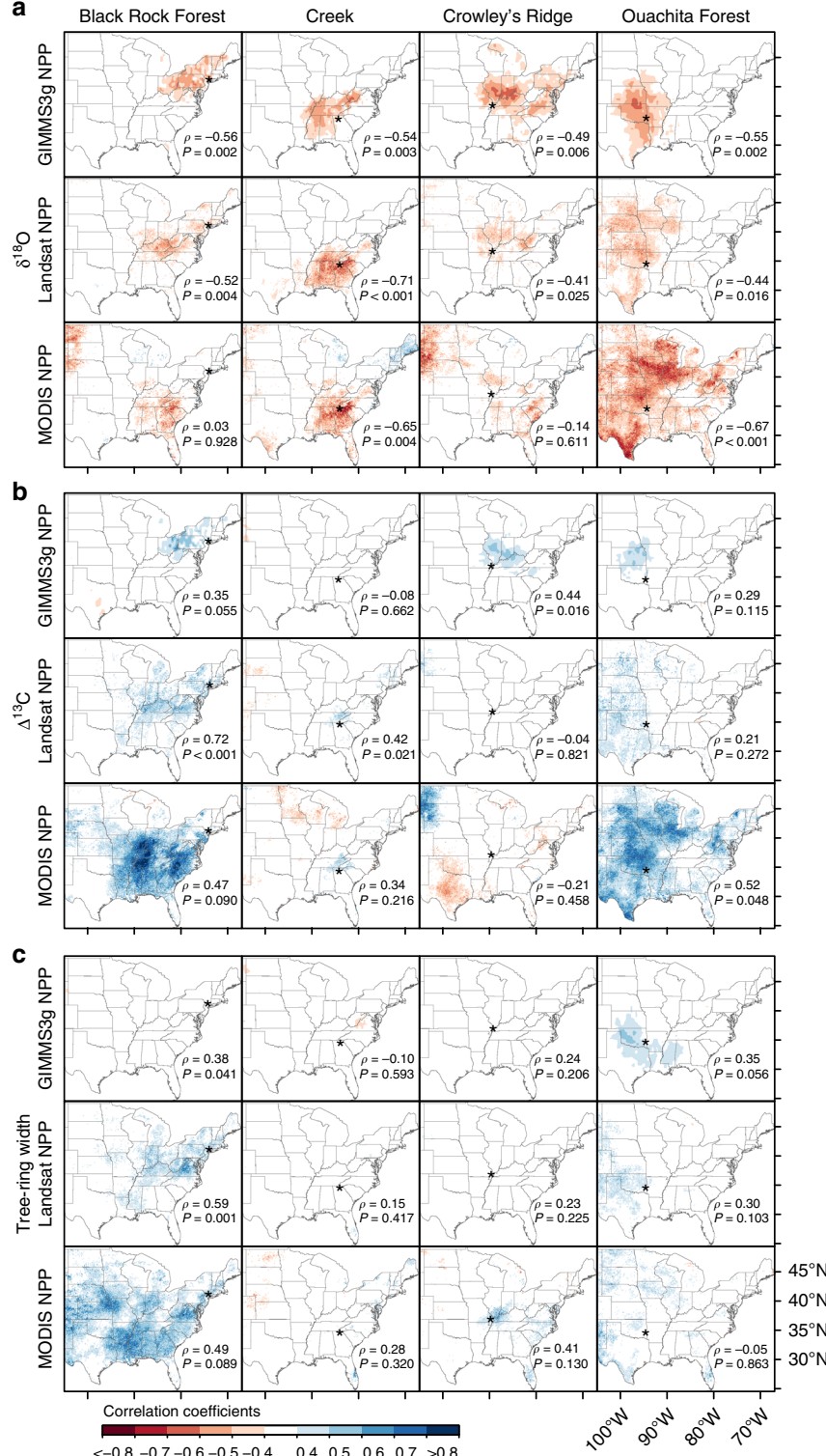

**Fig. 2** Spatial field correlations between tree-ring variables and satellite NPP products. Correlation coefficients were calculated between annually resolved tree-ring $\delta^{18}O$ (**a**), $\Delta^{13}C$ (**b**), and width (**c**) chronologies measured at four study sites and GIMMS3g NPP (1982–2011), Landsat NPP (1986–2015) and MODIS NPP (2001–2015) datasets. Study sites are represented by stars. Site-level correlations and their significance are indicated by Spearman's rank correlation coefficient ($\rho$) and $P$-value ($P$)

(Fig. 3a, b), and to a lesser extent $\Delta^{13}C$ data, indicate a strong coupling between moisture availability, tree physiology, and ecosystem productivity of the relatively moist US broadleaf forest biome.

Tree-ring width chronologies, in contrast, do not seem to consistently reflect remotely sensed NPP as stable isotope data. The weakest relation between tree-ring widths and remotely-sensed NPP occurs in the southern portion of our network (Frick

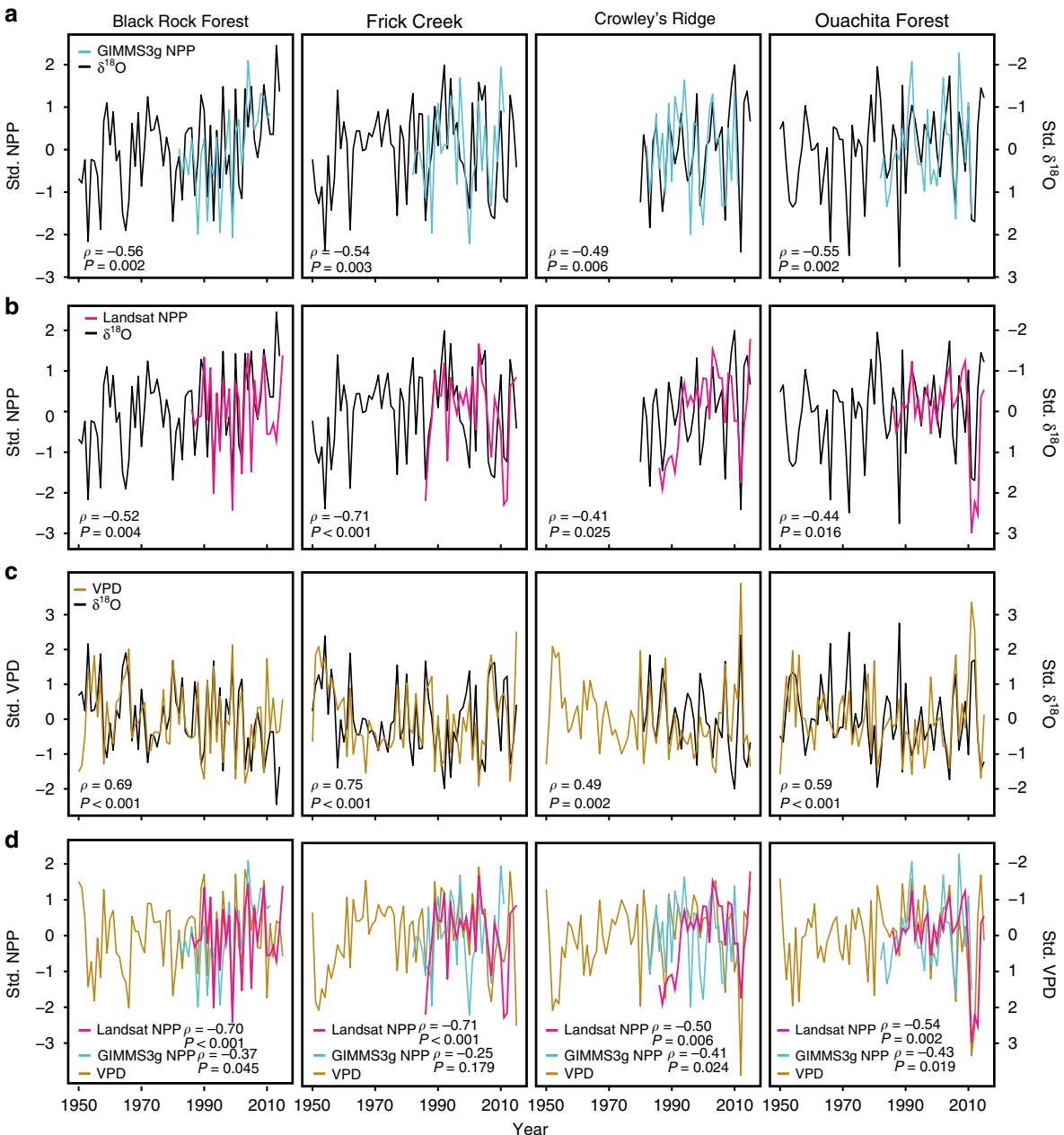

**Fig. 3** Tree-ring δ[18]O, GIMMS3g NPP, Landsat NPP, and vapor pressure deficit (VPD) time series at the study sites. Time series were standardized (Std.) and correlated to each other for the overlapping period. Tree-ring δ[18]O plotted against GIMMS3g NPP (**a**), Landsat NPP (**b**) and summer (June–August) VPD (**c**). VPD plotted against GIMMS3g NPP and Landsat NPP (**d**). Correlations and their significance are indicated by Spearman's rank correlation coefficient (ρ) and P-value (P). Note the inverse y-axis for Std. δ[18]O and VPD in **a**, **b**, and **d**

Creek), high in the southern Appalachian Mountains (Fig. 2c). While there is a trend towards reduced precipitation in this region, this particular area receives nearly temperate-rainforest levels of precipitation. Prior work indicates that higher daytime temperatures constrain growth more in this region than further north[17]. Thus, woody increment of trees at Frick Creek might be less responsive to cooling and very wet local conditions. In contrast to our southern collections, tree-ring widths in the northeastern US (Black Rock Forest), and to a lesser extent towards the western end of the US broadleaf forest biome (Ouachita Forest), have stronger relations to remotely-sensed NPP (Fig. 2c). The strongly significant NPP signal recorded by the tree-ring width data at Black Rock Forest might be partially related to shallow soils upon which the *Quercus rubra* trees grew

and the strong sensitivity of those trees to spring and summer VPD[11]. Overall, our results suggest that unlike Δ[13]C and δ[18]O, which agree very well with NPP data across large geographical areas, tree-ring width likely captures more local environmental conditions (e.g., site moisture, stand density, competition, stand dynamics). Also, since lags between carbon allocation, stem-girth increment (i.e., xylem cell production and enlargement), and woody biomass production (i.e., cell wall thickening and lignification) exist[31], mismatch between wood production (i.e., tree-ring width) and eddy covariance or satellite productivity observations might be expected[21,32].

In contrast to tree-ring width, Δ[13]C and δ[18]O measured in latewood cellulose record the conditions experienced at the canopy level such as evapotranspiration due to evaporative

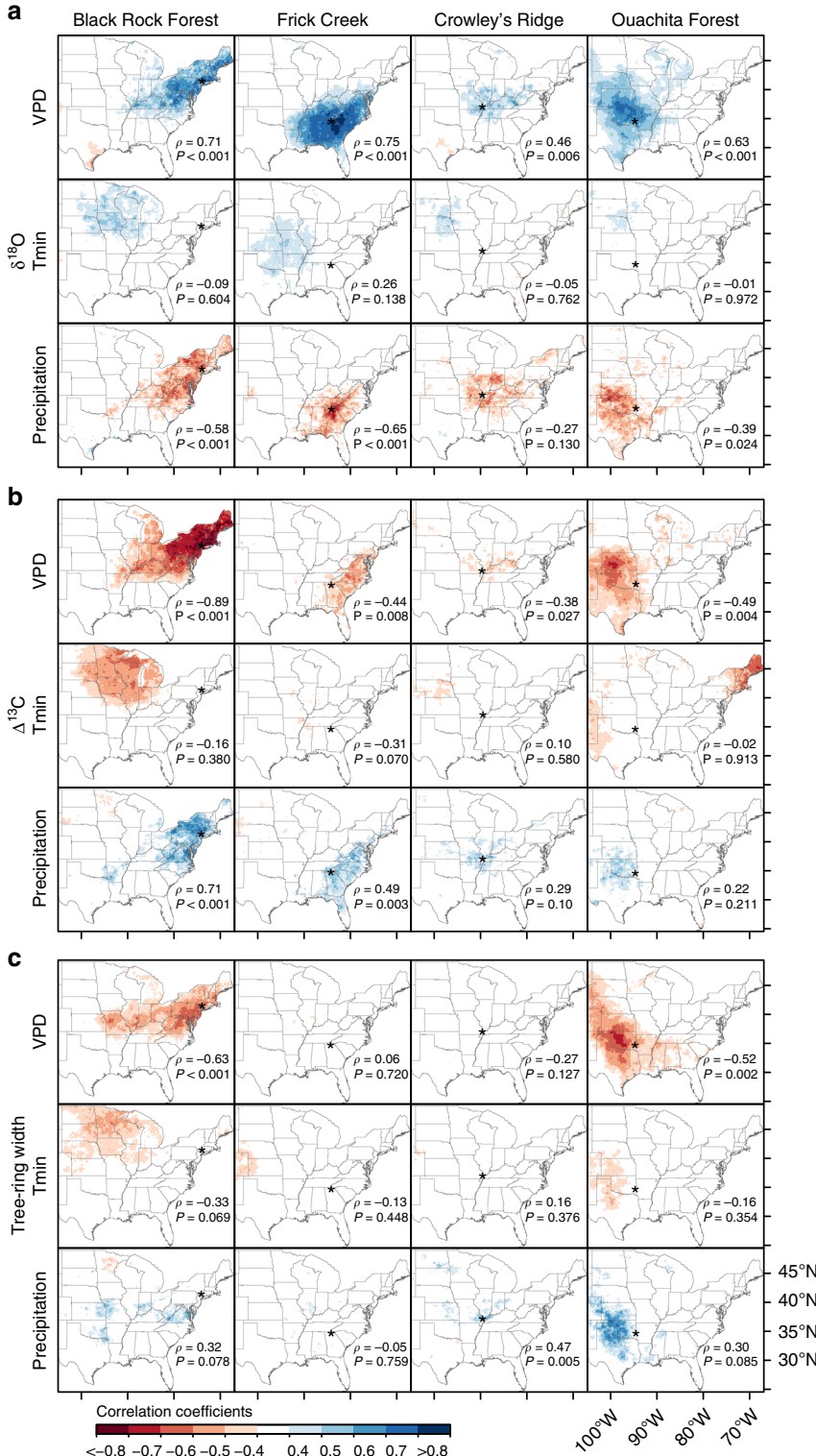

**Fig. 4** Spatial field correlations between tree-ring variables and gridded climate data. Correlation coefficients for the period 1982–2015 were calculated between annually resolved tree-ring δ18O (**a**), Δ13C (**b**), and width (**c**) chronologies measured at four sites and daytime mean May–August vapor pressure deficit (VPD), minimum temperature (Tmin), and precipitation. Study sites are represented by stars. Site-level correlations and their significance are indicated by Spearman's rank correlation coefficient (ρ) and P-value (P)

demand of the atmosphere (VPD), irradiance, and temperature at the time of the year when the wood was formed[25]. These environmental changes at the canopy level alter the LAI and FPAR, factors that are measured from the satellite sensors and needed for NPP calculations. Therefore, this link could potentially reinforce the association between isotopic chronologies from latewood and remotely sensed NPP. Still, our results indicate that most of the NPP signature recorded in the tree-ring isotopic data is related with the climatic variables, especially VPD, rather than the FPAR and LAI canopy signatures.

Several studies have explored the potential of tree-ring variables (e.g., widths, δ13C, maximum latewood density) for inferring vegetation productivity. Although first attempts to relate tree-ring width to productivity estimates from flux towers were rather inconclusive[32], a recent study reported a good agreement between flux tower productivity measurements and tree-ring δ13C of *Quercus rubra* and *Tsuga canadensis* in a northeastern US temperate forest[23]. Large-scale coherence was also reported between Normalized Difference Vegetation Index (NDVI) estimates and tree-ring width[33,34] and maximum latewood density data from tree-ring chronologies in the North American Boreal Forest[35] and Siberia[36]. Supporting these previous findings, our results indicate that tree-ring isotopes could complement space-based and land-based NPP estimates by providing high-resolution NPP reconstructions at annual to centennial time scales. Notably, our work shows how site-level Δ13C and δ18O tree-ring chronologies based on only five trees per species yield the potential to reconstruct past NPP across large geographical regions. Given that satellite estimates of NPP cover only the period 1982 to present and that repeated forest inventories have multi-annual to decadal gaps between consecutive measurements, tree-ring isotopes show high potential for providing high-resolution vegetation productivity estimates.

Although at this moment a formal reconstruction of NPP cannot be performed due to the short time spans of overlap between the tree-ring isotopic data and NPP time series (30 years for GIMMS3g and Landsat, 16 years for MODIS), this problem will reduce with time. An alternative approach to increase the amount of variance accounted by the tree-ring isotopic data needed for a statistically reliable reconstruction of NPP would be to generate more isotope tree-ring chronologies across the region through new sampling. There is precedence that increasing the diversity of species used for reconstructing past moisture availability improved the skill of paleoclimatic reconstructions[37,38]. Given that our isotopic data respond to moisture availability, we hypothesize that a similar kind of gain can be made as isotopic sampling increases across space and species in this region. This new approach could allow the assessment of vegetation growth trends to global change and the spatiotemporal monitoring of terrestrial carbon stocks at the regional to biome scale, and ultimately yield crucial information for the improvement of earth and biochemical models.

Previous studies comparing satellite estimates of vegetation productivity with forest inventories have found relatively good agreement between space-based and ground-based methodologies[39,40]. The main advantages of satellite-based NPP estimates over terrestrial inventories and flux tower data are their global coverage and remarkable accuracy at coarse scales[41]. Mismatches, at local or regional scales, however, were reported due to coarse resolution of the climate data[40], broad vegetation type classification, and underrepresentation of the soil water conditions[41] and stand density[39]. These limitations at local scales may be partly overcome by integrating tree-ring isotope data. At the tree level, δ13C and δ18O integrate abiotic (climate and site conditions) and biotic (ecophysiological response) factors[11], which influence vegetation productivity at local to regional scales. The inter-annual variability in tree-ring δ18O, for example, is directly related to evaporative demand of the atmosphere or VPD and site water availability[11], two important climatic drivers of NPP at global scale[2,26,42]. The δ18O-NPP agreement identified in our study could be used as a quantitative proxy for reconstructing past terrestrial productivity and thus, for constraining dynamic global vegetation models that fail to accurately capture inter-annual variability of carbon fluxes and long-term trends[1].

Process-based models like the MOD17 (see Methods) and the underlying productivity products used in our study have been widely used in environmental sciences for estimating and monitoring the spatiotemporal variability of terrestrial ecosystem productivity[1,2,26,43]. Until recently, the coarse input meteorological datasets and biome-level parameters used in the algorithm were well-known to reduce the reliability and spatial applicability of the remotely sensed productivity estimates[2,27]. To overcome these limitations, the new MODIS and Landsat datasets use high-resolution land cover classification and improved meteorological data[27]. These latest generation datasets provide the most observationally based productivity estimates available for the conterminous United States. The parameters used for the calculation of GPP and subsequently NPP were validated and calibrated with GPP estimates from 43 eddy covariance flux towers[27]. This is a well-established procedure and widely used method of validation[26,27,44,45]. This approach is used for validation because field estimates of NPP are not widely available. It is important to note that even when there are field estimates of NPP available, there is still typically a lot of bias/error associated with field based NPP data[43]. The newest GPP calculated with the optimized cross-validated parameters show strong improvement compared with those calculated with the original MOD17 parameters. Critical to our work, these latest productivity datasets perform very well over the US deciduous broadleaf forest when compared with flux tower data (Supplementary Fig. 5)[27].

Despite the noted improvements of the space-based NPP products used here, there are still some assumptions and limitations to be aware of. First, the MOD17 algorithm provides estimates of total NPP based on light use efficiency logic and uses simple ratios of aboveground to belowground partitioning at the biome scale. The difficulty to measure belowground NPP and the paucity of aboveground NPP data make the calibration of satellite-based NPP data with field-based NPP measurements challenging. Still, there is evidence that roughly equal allocation of NPP between canopy, wood, and fine root growth exists, and these components are linearly related to total NPP[43,46–48]. Second, respiration represents an important source of uncertainty in the NPP algorithm because it uses coarse resolution biome level allometric relationships and is calculated separately from GPP[27]. A commonly used method is to simply assume NPP is a fixed proportion of GPP[49]. In fact, there is the need for campaigns aimed at accurately estimating seasonal NPP dynamics in the field. As new field-based NPP and respiration data become available for calibration, the accuracy of remotely sensed NPP data will improve. Despite these limitations, our new approach relating tree-ring variables to the latest generation of remotely sensed productivity data could be seen as an independent, truly observational indicator of NPP.

We note that before tree-ring isotopic time series can be used for reconstructing NPP with confidence in time and over space, some limitations should be overcome. First, tree-ring isotope data are only mechanistically linked to local NPP, and the correlations with NPP across a wider region are representative of the large footprint of NPP-relevant climate variability. The development of denser tree-ring isotope network should be pursued in regions where climate variability is more spatially heterogeneous. Second, the small temporal overlap (14–34 years) between tree-ring time series and satellite NPP products reduces confidence in the statistical relationships upon which long-temporal reconstructions would be based. However, it is expected that this limitation will be less in the future as continuous satellite data records are extended.

In conclusion, in an era of rapidly changing environmental conditions, high-resolution, long-term estimates of vegetation productivity are important for monitoring ecosystem carbon stocks and fluxes across large spatial expanses. The originality of our finding is that tree-ring isotopic data from temperate trees appear to have the potential to reconstruct NPP across large

spatial scales and long periods of time. Reconstructions of historical NPP for decades or centuries across large areas would represent a significant advancement in understanding the drivers of NPP and their interactions, identifying historical climatic events that resulted in large terrestrial carbon fluxes, and ultimately improving parameterization of global carbon models. The big advantage of tree-ring based NPP estimates would be capturing past variations in productivity at annual, decadal, and multidecadal time periods, which NPP satellite estimates and flux tower measurements are unable to record given the short nature of these records. Our study opens the door to the development of further dendro-isotope network outside the eastern US temperate regions to expand the spatial coverage of NPP signature recorded in tree-ring isotopes.

## Methods

**Study sites and tree-ring data.** We developed tree-ring width and isotopic ($\delta^{13}C$, $\delta^{18}O$) chronologies from *Liriodendron tulipifera* L. and *Quercus rubra* L., two ecologically important and abundant tree species with a wide distribution range in eastern North America (Supplementary Fig. 1)[28] and contrasting physiological behavior (isohydric and anisohydric, respectively)[50]. We selected four sites located in protected and unmanaged forests, close to the species distribution range and representing three distinct hydroclimatic conditions. We sampled trees from the dry and hot forest-prairie ecotone in Oklahoma (dry western edge of the broadleaf forest biome), the wet and warm southern Appalachian Mountains, and the moist and cool Northeastern US (Fig. 1, Supplementary Table 1). We sampled both *Liriodendron tulipifera* and *Quercus rubra* at Black Rock Forest NY (41°24′N and 74°01′W) and in the Chattahoochee National Forest GA (site Frick Creek, 34°40′ and 84°11′W). Due to differences in their western range margins, *Liriodendron tulipifera* was sampled at General Watkins Conservation Area MO (site Crowley's Ridge, 37°04′N and 89°36′W) and only *Quercus rubra* was sampled in the Ouachita National Forest OK (34°41′N and 94°38′W).

At each site, we sampled 15–20 mature and dominant trees with full canopies to minimize potential stand competition and ontogenetic effects on tree-ring isotopic ratios[51] and took two 5 mm diameter increment cores from each tree (Supplementary Table 2). The increment cores were air dried, glued on wood mounts, and successively sanded with finer grades of sandpaper until the xylem structure and ring boundaries were clearly visible. Ring widths from increment cores were visually crossdated and measured to the nearest 0.001 mm. Visual crossdated measurements of ring width were then statistically tested using the software COFECHA[52]. Each individual ring-width series was detrended with a cubic smoothing spline with a 50% frequency cutoff equal to two-thirds the series length to remove non-climatic trends, and a first-order autoregressive model was fitted. Detrended and residual individual series were averaged with a biweigthed robust mean to build a master chronology for each species at each site[53]. To ensure that the number of trees sampled was sufficient and representative of the sampled population, we calculated the Express Population Signal (EPS). All tree-ring chronologies showed EPS values ≥ 0.85 (Supplementary Table 2), which is considered the threshold value for adequately reflecting a common signal among trees[54].

For isotopic analysis, we selected the five trees per species with the highest correlations with the tree-ring width master chronology and used an extra 5 or 12 mm core (Supplementary Table 3). For each tree, we analyzed the isotopic ratios for each annual ring individually for the periods: 1950–2015 (Frick Creek and Ouachita Forest sites), 1950–2014 (Black Rock Forest site), and 1980–2015 (Crowley's Ridge site). We set the beginning of the investigation period (i.e., 1950 or 1980) to focus the analysis at the time when the trees were already in the upper canopy and had nearly reached their maximum height to minimize the potential effect of changes in tree height on isotopic ratios[51]. From each core, we split off the latewood of each annual ring with a scalpel under a stereomicroscope, chopped the material, and stored each wood sample individually in centrifugal tubes before cellulose extraction. Latewood is less dependent on stored carbon and evaporative enrichment that occurred during the previous growth year and therefore corresponds to the isotopic signal of recently assimilated C and O during the growing season[55]. *Liriodendron tulipifera* does not have a distinguishable latewood section. In that case, we analyzed the last third of the ring. Thus, we analyzed the $\delta^{13}C$ and $\delta^{18}O$ ratios in α-cellulose from tree-ring latewood or the most recent portion of each ring (hereafter referred as latewood) instead of the entire ring. We extracted the α-cellulose following standard procedures described in [56] and homogenized the cellulose using an ultrasound treatment[57]. Dendrochronological and geochemical proceedings were conducted at the Tree-Ring Lab and at the Terrestrial Ecology Lab, respectively, of the Lamont-Doherty Earth Observatory of Columbia University. For each sample, 200 μg of cellulose were weighted and put in silver capsules. $\delta^{13}C$ and $\delta^{18}O$ ratios were measured simultaneously using high-temperature pyrolysis in a Costech elemental analyzer interfaced with an Elementar Isoprime mass spectrometer at the Department of Geology at the University of Maryland, USA[58]. The analytical precision for the in-house α-cellulose standards

was ±0.12‰ for $\delta^{13}C$ and ±0.28‰ for $\delta^{18}O$. To correct for anthropogenic changes in $\delta^{13}C$ of atmospheric $CO_2$, tree-ring $\delta^{13}C$ values were used to calculate carbon isotope discrimination values ($\Delta^{13}C$) and were defined as the δ value measured in tree-ring cellulose ($\delta^{13}C_{trc}$) against that of atmospheric $CO_2$ ($\delta^{13}C_{CO2}$) using Equation (1)

$$\Delta^{13}C = \delta^{13}C_{CO2} - \delta^{13}C_{trc} / \left(1 + \frac{\delta^{13}C_{trc}}{1000}\right) \quad (1)$$

**Satellite NPP datasets.** We used the latest 30-m resolution Landsat and 250-m resolution MODIS NPP products for the conterminous United States from the Numerical Terradynamic Simulation Group (http://www.ntsg.umt.edu/project/landsat/landsat-productivity.php)[27]. Both datasets are calculated based on the MODIS MOD17 algorithm by using satellite observations of the absorbed fraction of photosynthetically active radiation (FPAR) and leaf area index (LAI), land cover classification, as well as daily 4-km gridded solar radiation, temperature, and vapor pressure deficit (VPD) data from the University of Idaho's METDATA[59]. We resampled the MODIS and Landsat NPP datasets at 1-km resolution and used the overlapping period between the tree-ring chronologies and yearly Landsat (1986–2015) and MODIS (2001–2015) NPP time series for the analysis. In our analysis, we also used the 10-km resolution annual GIMMS3g NPP dataset[1]. The GIMMS3g NPP dataset was also calculated using the MOD17 algorithm but used the GIMMS3g FPAR and LAI as input data[60] from the Advanced Very High Resolution Radiometer (AVHRR) for the period 1982–2011[1].

**MOD17 algorithm.** The MOD17 algorithm directly relates gross primary productivity (GPP) and total NPP to the amount of solar radiation absorbed by the plant canopy[61]. It uses four main variables: the FPAR, LAI, meteorological measurements (temperature, VPD) and land cover classification. The algorithm calculates GPP as

$$GPP = LUE_m \times f_{Tmin} \times f_{VPD} \times SW_{rad} \times 0.45 \times FPAR, \quad (2)$$

where $LUE_{max}$ (g C MJ$^{-1}$) is a biome specific maximum light use efficiency multiplied by minimum temperature ($f_{Tmin}$) and vapor pressure deficit ($f_{VPD}$) scalars to account for low temperature limits and water stress, respectively. $SW_{rad}$ (w m$^{-2}$) is the incoming shortwave radiation of which we assume 45% is in the wavelengths available for photosynthesis. FPAR (unitless) corresponds to the estimated fraction of photosynthetically active radiation captured by the plant canopy. Annual NPP is calculated from GPP after accounting for maintenance ($R_M$) and growth respiration ($R_G$) as

$$NPP = \sum_{i=1}^{365}(GPP_i - R_{M_i}) - R_G = \sum_{i=1}^{365}(GPP_i - R_{M_i}) - 0.25 \times NPP, \quad (3)$$

where $R_M$ (g C m$^{-2}$ d$^{-1}$) is calculated using remotely sensed values of LAI (m$^2$ leaf m$^{-2}$ ground), biome specific parameters and climate data. Allometric relationships between estimated leaf area, leaf mass, fine root mass and live wood mass are used to estimate $R_M$. Growth respiration is assumed to be approximately 25% of NPP[2,62]. All biome parameters used in the MOD17 algorithm were calibrated for the conterminous United States with eddy covariance flux tower data and are reported in the updated biome properties lookup table in Robinson et al.[27].

**Statistical analysis.** Spearman's rank correlation analysis was used to assess the agreement shared between site-level tree-ring width, $\Delta^{13}C$, and $\delta^{18}O$ chronologies and local NPP time series extracted from the grid cell nearest each study site. We further calculated and mapped the spatial extent of the interannual NPP signal recorded in tree-ring chronologies using spatial field correlation analysis. Specifically, correlation coefficients were calculated between the tree-ring width, $\Delta^{13}C$, and $\delta^{18}O$ chronologies from each study site and each grid cell of the Landsat NPP, MODIS NPP and GIMMS3g NPP raster time series for the eastern US. Correlation coefficients calculated for each grid cell that were above 0.40 and below −0.40 were mapped. To assess the potential FPAR and LAI signals contained in tree-ring data, we correlated the tree-ring chronologies with the FPAR and LAI raster time series from the MODIS FPAR and LAI products (period 2000–2015) and GIMMS3g FPAR3g and LAI3g dataset (period 1982–2011). We also tested the sensitivity of the tree-ring data to daytime vapor pressure deficit (VPD), and minimum temperature, two climatic variables used to constrain GPP, and precipitation with correlation analysis. Temperature, precipitation, and VPD 4-km resolution gridded data were obtained from the PRISM Climate Group, Oregon State University (http://prism.oregonstate.edu). For the analysis, we combined and averaged the tree-ring chronologies of *Liriodendron tulipifera* and *Quercus rubra* for Black Rock Forest and Frick Creek, whereas we used the chronologies of *Liriodendron tulipifera* for Crowley's Ridge and *Quercus rubra* for Ouachita Forest.

## Data availability

All tree-ring data used in this publication are available as a supplementary file (Supplementary Data 1). Landsat and MODIS NPP data are available from the

Numerical Terradynamic Simulation Group (http://www.ntsg.umt.edu/project/landsat/landsat-productivity.php). GIMMS3g NPP data can be downloaded here (https://wkolby.org/data-code/). Additional data related to this paper may be requested from the authors.

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

## Acknowledgements
We thank William Schuster and the Black Rock Forest Consortium, Nelson E. Gonzalez-Sürrow (Chattahoochee-Oconee National Forest), Norman Wagoner (Ouachita National Forest), the USDA Forest Service and Malissa Briggler (Missouri Department of Conservation) for sampling authorization, as well as Javier Martin-Fernandez and Richard Li for their help in the field and with sample preparation. Thank you to Jonathan Thompson and Matthew Duveneck whose stimulating thoughts played a role in developing this research. This work was funded by a Lamont-Doherty Earth Observatory Climate Center grant and by the National Science Foundation grant no. PLR 15–04134. M.L. was supported by an Early and Advanced Postdoc Mobility Fellowships from the Swiss National Science Foundation (project numbers: P2EZP2_152213 and P300P2_164637). L.A.H. and A.P.W. were supported by Columbia University's Center for Climate and Life. Lamont-Doherty Earth Observatory contribution no. 8724.

## Author contributions
M.L., L.A.H., A.P.W., and N.P designed the study. M.L. conducted fieldwork, performed and supervised tree-ring sample preparation and isotopic measurements, analyzed the data, and wrote the manuscript. W.K.S. provided GIMMS3g NPP, FPAR3g, and LAI3g remote sensing data. M.L.H. provided MODIS FPAR and LAI remote sensing data. B.W.A. calculated and provided Landsat and MODIS NPP satellite products. All authors contributed to the interpretation of the results and final version of the manuscript.

## Additional information

**Competing interests:** The authors declare no competing interests.

