## [Peer Review File · Nature Communications]

Reviewers' comments:

Reviewer #1 (Remarks to the Author):

This manuscript describes the results of a study describing the relationships among retrospective satellite-derived productivity estimates, tree-ring stable isotope ratios, and tree-ring widths. The success of such a search for correlation depends on the variability in the data analyzed. In this case, the study begins with a comparison of tree rings from two tree species from four individual sites. The sites were located near the climatic limits of the temperate deciduous forests of North America. It then correlates the tree-ring data at these sites to satellite estimates of productivity over the whole region, mapping regions of strongest correlation.

As an ecophysiologicalist, I am always glad to see stable isotopes mixed into a correlation analysis. It's a clever way of introducing just a bit of physiology into the exercise. That said, it's an easy thing to do, and will not necessarily lead to great insight. It all depends on how the analysis is done and how the data are interpreted.

In this case, the paper needs a better justification and more careful description of the sample design before we can determine what the results mean. First, why were these particular sites chosen for the analysis? Second, why these particular species? They are used as if they were indicators of the other major species at the site. Is there a compelling reason to believe that? Third, we need more information about the individual trees at the sites. This should include how they were chosen. In addition, we need to know the height, diameter, and stand density if we are to assess the risk of hidden sources of variability within the sample population. In particular I am concerned that older tall trees are being compared to younger short trees (see the recent paper by (Brienen *et al.*, 2017). If so, this needs to be made clear.

I also found myself wishing for further description of the reliability of the NPP data. It's important to know where the numbers come from to be able to know how much faith to put in them. They are derived from light capture and photosynthesis, with little additional constraint by empirical data. Are the data in the Midwest for cornfields? If so, one must question the value of correlations with forest tree parameters. Have the satellite estimates of NPP been ground-truthed near any of these sites? Has it been done with an independent dataset anywhere? I presume so, but it is up to the authors to present this comparison.

Much is made of the strength of the correlations. But they explain about 25% of the variance at best. It's hard for me to see that as strong. But what is strong and what is weak? Perhaps it would be better to compare these predictions to ones that did not include isotopic data or to previous reports of satellite production vs. ground truth. It seems that one of the main purposes of the manuscript is to use stable isotopes to improve these predictions. If so, then we need some demonstration that it worked better than some earlier alternative. This is also true of the regional estimates, which could also be compared to models lacking stable isotopes.

Brienen, R. J. W., Gloor, E., Clerici, S., Newton, R., Arppe, L., Boom, A., ... Timonen, M. (2017). Tree height strongly affects estimates of water-use efficiency responses to climate and CO₂ using isotopes. *Nature Communications*, 8(1), 288. <https://doi.org/10.1038/s41467-017-00225-z>

Reviewer #2 (Remarks to the Author):

Review

Tree-ring isotopes reflect satellite-based net primary productivity estimates at the biome scale
Submitted by Levesque et al. to *Nature Communications*

SUMMARY

This is an interesting and innovative study addressing a scientifically relevant objective. The demonstration of the ability to reconstruct NPP estimates over longer timescales opens opportunities for a suite of new applications that will likely increase our understanding of pre-Anthropocene productivity changes and eventually support a clearer (less uncertain) perspective on upcoming decades dominated by greenhouse gases and climate change.

The detailed statistical associations between tree-ring stable isotope chronologies from ecologically differing sites in the eastern United States and satellite derived NPP estimates appear to be robust, and the fractions of explained variance are considerable, so that proxy-based reconstructions of this important component of the carbon budget appear foreseeable.

In this respect, the study by Levesque et al. provides a novel approach that will likely stimulate subsequent work further exploring this relationship and applying stable isotope-NPP relationships to formal reconstructions. Since the study is also technically sound, and the manuscript well written, only minor revisions are recommended.

The statement on Data and materials availability appears unsatisfying though ("Additional data...may be requested from the authors.") Clearly, the manuscript presents important and original stable isotope data that should be made available online, e.g. in the International Tree Ring Data Bank. This seems to be important.

DETAILED COMMENTS

Its not fully clear to me, why the comparison with MODIS NPP extends until 2015, whereas the correlations with GIMMS3g include data until 2011 only. Is this because the latter terminated that year (I am seemingly not an expert on this, but found a hint for 2012 in the www)? This should particularly be stated to avoid confusion.

Lines 43-44: "...at both local and large geographical scales..."

Line 57: Consider removing "and trajectories"

Line 62: Consider removing "estimate"

Line 76: Reference 11: This is a review paper. Perhaps better use an application paper, such as "Frank DC, Poulter B, Saurer M, Esper J, et al. (2015) Water use efficiency and transpiration across European forests during the Anthropocene. Nature Climate Change 5, 579-583."

Line 80: Consider removing sentence starting with "It is also....", as this might very generally be the case for many applications/targets.

Line 89: Again, replace reference 11 by a case study.

Line 93: Remove "for the first time".

Line 97: Perhaps state more clearly/with more detail what is meant with "contrasting hydroclimatic trends".

Line 104, and thereafter: Not sure about the delta sign for the stable carbon isotopes. Should this be uppercase (d), not lowercase?

Lines 106ff: Could the correlation even be increased by considering the significantly correlating field surrounding the tree sites? If so, this would be worth to mentioned.

Line 127: Remove "tree ring based" and "broadly".

Line 149: Replace "and significantly to" with "with".

The paragraph from lines 143-157 contains slightly too many correlations values. It seems worth shortening this to increase readability.

I like the figures and believe that they are all worth to be included in the main text -- except for figure 5, which is fairly empty (for good reason) and could be moved to the supplement.

Lines 173-180: This first Discussion paragraph could be moved to the conclusions (or be deleted, re-worked).

Line 194: "...by the atmospheric evaporative demand..."

Figure 1: Perhaps zoom-in a bit (e.g. truncate at 100W and 45N) to show more detail in the vicinity of the sampling sites.

Figure 2, 4 and 5: The statement on the combination of Liriodendron and Quercus chronologies (and so on...) is repeated in three legends. Perhaps state this once in the Methods section and remove from the legends.

Reviewer #3 (Remarks to the Author):

General comment:

To have a reliable long-term record of NPP is vital to understand the impact of climate trend and variability not only on forest carbon stock but also on global terrestrial carbon cycle. This study found significant correlation between tree $\delta^{18}O$ and satellite based NPP estimate over multiple sites in eastern US forests, being stronger than the correlation between tree ring width and satellite NPP. This finding is encouraging as it provides a readily avenue for reconstructing long-term variations in forest NPP and hence tracing back the climate impact to a longer historical domain beyond the era of eddy-covariance and satellite techniques. These reconstructed long-term NPP data record, therefore, can be used to test dynamic global vegetation models for their ability to simulate observed productivity-climate relationship. As such, I support this study to be accepted for publication in Nature Communications with a minor revision.

The paper is well written and the research design is sound. I have no major concern on methodology implemented and

hence the results generated. My only concern is that this study only compared tree ring isotope measurements with satellite-based estimates of NPP but not in-situ NPP data, for instance, from forest inventories which can serve as ground-truth for validation purpose. To gain the readers the confidence of the use of MOD17 / GIMMS3g NPP product in this study, the author may refer to the performance of MOD17 NPP product over temperate forests (e.g., if there is any previous validation effort on space-based NPP estimates).

Specific comment

L71: it is correct that we need long-term NPP proxy. Besides the method the authors developed in this study, I am just curious to know are there any other proxie (besides tree-ring width) that have been developed and tested? If so, the authors should review them and discuss the differences with the proxy they developed;

L127: the weaker correlation between tree-ring width and satellite NPP may indicate a lag response of wood productivity and annual NPP? not sure if would be worth to test lagged correlation between NPP and tree-width time series?

L342: the author may indicate which version of MOD17 product they used in this study (C5 or C6).

Fig. 1: the value appears to be too high for the slope of trend in T or P. What is the unit of trend? per year or per decade?

Fig. 3: maybe the author can also show time series of MOD17 NPP?

Reviewer #1 (Remarks to the Author):

This manuscript describes the results of a study describing the relationships among retrospective satellite-derived productivity estimates, tree-ring stable isotope ratios, and tree-ring widths. The success of such a search for correlation depends on the variability in the data analyzed. In this case, the study begins with a comparison of tree rings from two tree species from four individual sites. The sites were located near the climatic limits of the temperate deciduous forests of North America. It then correlates the tree-ring data at these sites to satellite estimates of productivity over the whole region, mapping regions of strongest correlation.

As an ecophysiologicalist, I am always glad to see stable isotopes mixed into a correlation analysis. It's a clever way of introducing just a bit of physiology into the exercise. That said, it's an easy thing to do, and will not necessarily lead to great insight. It all depends on how the analysis is done and how the data are interpreted.

In this case, the paper needs a better justification and more careful description of the sample design before we can determine what the results mean. First, why were these particular sites chosen for the analysis? Second, why these particular species? They are used as if they were indicators of the other major species at the site. Is there a compelling reason to believe that? Third, we need more information about the individual trees at the sites. This should include how they were chosen. In addition, we need to know the height, diameter, and stand density if we are to assess the risk of hidden sources of variability within the sample population. In particular I am concerned that older tall trees are being compared to younger short trees (see the recent paper by (Brienen et al., 2017)). If so, this needs to be made clear.

Thank you very much for your constructive comments. We have added information justifying our choice of sites and species in the text (lines 101-107, 309-320). Basically, we targeted these species because of their abundance and ecological importance, their wide distribution range in eastern North America, and their contrasting physiological behavior (isohydric vs anisohydric). We selected sites close to the northern, eastern and western distribution limits of the species with contrasting climatic conditions for covering the largest environmental gradient possible.

We have added a new table with information on diameter, height, and age of the trees sampled for isotopic analysis (new Supplementary Table 3) and explained in the manuscript that only mature and dominant trees were sampled to minimize the potential height effect on tree-ring isotopic ratios (lines 322-324, 341-343) as reported by Brienen et al. (2017).

Further, for the trees that we sampled at Black Rock Forest, Frick Creek and Ouachita Forest, we limited our isotopic analysis for the period 1950-2015 to minimize the potential height effect on tree-ring isotopic ratios. According to site index curves for those sites, the sampled trees had nearly reached their maximum height in 1950 as it is estimated that tree height has only increased by 3-5 m in the following 65 years. For the same reasons and because trees sampled at Crowley's Ridge were younger, we limited the isotopic analysis for the period 1980-2015. Therefore, with this sampling strategy and the fact that we made more of an ecological collection than dendroclimatic studies of the past, we are confident that the potential height bias on isotopic ratios as reported by Brienen et al. (2017) was inexistent or very minimal in our study.

I also found myself wishing for further description of the reliability of the NPP data. It's important to know where the numbers come from to be able to know how much faith to put in them. They are derived from light capture and photosynthesis, with little additional constraint by empirical data. Are the data in the Midwest for cornfields? If so, one must question the value of correlations with

forest tree parameters. Have the satellite estimates of NPP been ground-truthed near any of these sites? Has it been done with an independent dataset anywhere? I presume so, but it is up to the authors to present this comparison.

These are good questions you are asking here. The satellite-based NPP data we use in our study have been calibrated with empirical data from flux towers across the globe (Smith *et al.*, 2016, Zhao & Running, 2010). The latest generations of high-resolution satellite NPP data (Landsat 30m and MODIS 250m) that we used were recently compared with flux tower data for deciduous broadleaf forests in eastern US and showed a very strong similarity (Robinson *et al.*, 2018). Those data have only become available in the past several weeks, so we re-ran the analyses with these new high-resolution datasets (Landsat and MODIS NPP) and have included the results in the revised version. Briefly, we found that our tree-ring isotopic chronologies match as good if not better the latest generation of satellite NPP data (Landsat and MODIS NPP) than the older MODIS NPP data used in the original submission.

We have also included the following figure comparing flux tower GPP and MODIS and Landsat GPP in the Supporting Information (new Supplementary Fig. 4).

Supplementary figure 4. MODIS gross primary production 250m (GPP_{M250}) and Landsat GPP 30m (GPP_{L30}) relative to GPP measured at flux towers (FLUXNET2015) from deciduous broadleaf forest for the conterminous US. From Robinson *et al.* (2018).

Much is made of the strength of the correlations. But they explain about 25% of the variance at best. It's hard for me to see that as strong. But what is strong and what is weak? Perhaps it would be better to compare these predictions to ones that did not include isotopic data or to previous reports of satellite production vs. ground truth. It seems that one of the main purposes of the manuscript is to use stable isotopes to improve these predictions. If so, then we need some demonstration that it worked better than some earlier alternative. This is also true of the regional estimates, which could also be compared to models lacking stable isotopes.

We agree that 25% of the variance is not yet up to standards for reconstructing NPP from tree-ring derived isotopes. Yet, this initial sampling is much stronger than expected given the small sample size and replication. That some collections capture the trend in the satellite derived NPP is promising (Fig. 3a). There is precedence that increasing the diversity of species used for reconstructing past moisture availability has been shown to increase the skill of paleoclimatic reconstructions (Maxwell *et al.*, 2011, Pederson *et al.*, 2013). Given that our isotopic data here are responding to moisture availability (VPD), we hypothesize that a similar kind of gain can be made as isotopic sampling increases across space and species in this region. Another limitation that will be circumvented in the future is that at this moment a formal reconstruction of NPP cannot be

performed due to the short time spans of overlap between the tree-ring isotopic data and NPP timeseries (30 years for GIMMS3g and Landsat, while just 16 years for MODIS). As years passed by estimates of satellite-derived NPP will get longer allowing for an increase of the time span of the calibration period in the future, and thus, it will be possible to generate a reliable reconstruction of past NPP variability.

In order to clarify this concern, new text (lines 249-259) has been added in the main manuscript .

Overall, the novelty of our study lies in the comparison of tree-ring data (width, carbon and oxygen isotopes) with high-resolution satellite NPP data. In our study, the isotopic data appear much better correlated with NPP than do tree radial growth (tree-ring width). Further, as outlined above, satellite productivity data match very closely those from flux tower (Fig. 1 above). We are thus confident that isotopic ratios measured in tree-ring cellulose has a good potential for inferring NPP at local and regional scales. Earlier studies comparing satellite products with tree-ring variables only focused on the normalized Difference Vegetation Index (NDVI) estimates and tree-ring width. Although these previous works found a relatively good agreement between NDVI and tree-ring width, it is important to note that NDVI is not a true measure of vegetation productivity. Our analysis used the newest generations of satellite NPP data that were calibrated and tested against ground-based productivity data from flux towers in different biomes.

Reviewer #2 (Remarks to the Author):

Review

Tree-ring isotopes reflect satellite-based net primary productivity estimates at the biome scale
Submitted by Levesque et al. to Nature Communications

SUMMARY

This is an interesting and innovative study addressing a scientifically relevant objective. The demonstration of the ability to reconstruct NPP estimates over longer timescales opens opportunities for a suite of new applications that will likely increase our understanding of pre-Anthropocene productivity changes and eventually support a clearer (less uncertain) perspective on upcoming decades dominated by greenhouse gases and climate change.

The detailed statistical associations between tree-ring stable isotope chronologies from ecologically differing sites in the eastern United States and satellite derived NPP estimates appear to be robust, and the fractions of explained variance are considerable, so that proxy-based reconstructions of this important component of the carbon budget appear foreseeable.

In this respect, the study by Levesque et al. provides a novel approach that will likely stimulate subsequent work further exploring this relationship and applying stable isotope-NPP relationships to formal reconstructions. Since the study is also technically sound, and the manuscript well written, only minor revisions are recommended.

The statement on Data and materials availability appears unsatisfying though ("Additional data...may be requested from the authors.") Clearly, the manuscript presents important and original stable isotope data that should be made available online, e.g. in the International Tree Ring Data Bank. This seems to be important.

Thank you for reviewing our manuscript and for the constructive comments. We have included the tree-ring data (width, carbon and oxygen isotopes) used in the analysis as online supplementary data.

DETAILED COMMENTS

Its not fully clear to me, why the comparison with MODIS NPP extends until 2015, whereas the correlations with GIMMS3g include data until 2011 only. Is this because the latter terminated that year (I am seemingly not an expert on this, but found a hint for 2012 in the www)? This should particularly be stated to avoid confusion.

We apologize for this confusion. The comparison with MODIS NPP extends until 2015 because our tree-ring data extend until 2015. The GIMMS3g NPP dataset, on the other hand, only covers the period 1982-2011 (Smith et al. (2016). We now clarified in the text that we have different periods of analysis depending on the availability of the satellite-based NPP datasets (lines 99-101, 375-381).

Lines 43-44: "...at both local and large geographical scales..."

Done

Line 57: Consider removing "and trajectories"

Done

Line 62: Consider removing "estimate"

Done

Line 76: Reference 11: This is a review paper. Perhaps better use an application paper, such as "Frank DC, Poulter B, Saurer M, Esper J, et al. (2015) Water use efficiency and transpiration across European forests during the Anthropocene. *Nature Climate Change* 5, 579-583."

Thank you, we now cite the paper you suggested.

Line 80: Consider removing sentence starting with "It is also....", as this might very generally be the case for many applications/targets.

Done

Line 89: Again, replace reference 11 by a case study.

Changed for: Farquhar, GD and RA Richards. "Isotopic Composition of Plant Carbon Correlates with Water-Use Efficiency of Wheat Genotypes." *Australian Journal of Plant Physiology* 11, no. 6 (1984): 539–52.

Line 93: Remove "for the first time".

Done

Line 97: Perhaps state more clearly/with more detail what is meant with "contrasting hydroclimatic trends".

We have added the following sentence (lines 104-107):

We sampled a site in the northeastern US, a region exhibiting strong wetting and mild warming trends and three sites located in the southeast and central US where warming has not yet taken place (Mascioli *et al.*, 2017), but where summer precipitation trends are spatially variable

Line 104, and thereafter: Not sure about the delta sign for the stable carbon isotopes. Should this be uppercase (δ), not lowercase?

We measured $\delta^{13}\text{C}$ in tree rings but used the carbon isotope discrimination ($\Delta^{13}\text{C}$) for analysis since it accounts for changes in atmospheric $\delta^{13}\text{C}$. Only $\Delta^{13}\text{C}$ is used in the analysis. $\Delta^{13}\text{C}$ express the rates of discrimination against the heavy isotope (^{13}C) depending on the amount of total CO_2 available at leaf level. Reduction on stomatal aperture and higher photosynthetic rates will lead to lower CO_2 concentration in stomatal chambers, that will yield to lower discrimination against the heavy isotope. We have provided the definition and equation in the Methods section (lines 359-364).

Lines 106ff: Could the correlation even be increased by considering the significantly correlating field surrounding the tree sites? If so, this would be worth to mentioned.

For our correlation analyses, it certainly would enhance the correlation coefficients we report if we were to relate our tree-ring records to records of NPP from only those grid cells where NPP correlates significantly with our tree-ring data. However, we believe that screening out locations where correlations are weak could be argued as a method that artificially inflates the correlation values and we therefore choose to retain our more conservative method: For local correlations, we extracted the satellite NPP data for the closest grid cell at each study site and correlated this NPP time series with our tree-ring width and isotope measurements. For regional correlations, we related the tree-ring records to mean NPP across the entire eastern US.

Line 127: Remove "tree ring based" and "broadly".
Done

Line 149: Replace "and significantly to" with "with".
Done

The paragraph from lines 143-157 contains slightly too many correlations values. It seems worth shortening this to increase readability.

We have rewritten this paragraph for clarity (lines 152-162).

I like the figures and believe that they are all worth to be included in the main text -- except for figure 5, which is fairly empty (for good reason) and could be moved to the supplement.

Thank you for this suggestion., We have moved figure 5 to supplementary information (new Supplementary Fig. 3).

Lines 173-180: This first Discussion paragraph could be moved to the conclusions (or be deleted, re-worked).

We absolutely agree, thank you. We have moved this paragraph to the conclusion section and rewritten this section (lines 291-298).

Line 194: "...by the atmospheric evaporative demand..."

Done

Figure 1: Perhaps zoom-in a bit (e.g. truncate at 100W and 45N) to show more detail in the vicinity of the sampling sites.

We appreciate the comment and without doubt this will make the plots slightly bigger. However, we think that it is important to use the same extent for all the maps to be consistent and facilitate the comparison between maps. For this reason, we have decided to keep the original extent.

Figure 2, 4 and 5: The statement on the combination of Liriodendron and Quercus chronologies (and so on...) is repeated in three legends. Perhaps state this once in the Methods section and remove from the legends.

Following your suggestion, we have moved this statement to the Methods section (lines 398-401) and removed it from the legends.

Reviewer #3 (Remarks to the Author):

General comment:

To have a reliable long-term record of NPP is vital to understand the impact of climate trend and variability not only on forest carbon stock but also on global terrestrial carbon cycle. This study found significant correlation between tree $\delta^{18}\text{O}$ and satellite based NPP estimate over multiple sites in eastern US forests, being stronger than the correlation between tree ring width and satellite NPP. This finding is encouraging as it provides a readily avenue for reconstructing long-term variations in forest NPP and hence tracing back the climate impact to a longer historical domain beyond the era of eddy-covariance and satellite techniques. These reconstructed long-term NPP data record, therefore, can be used to test dynamic global vegetation models for their ability to simulate observed productivity-climate relationship. As such, I support this study to be accepted for publication in Nature Communications with a minor revision.

The paper is well written and the research design is sound. I have no major concern on methodology implemented and hence the results generated. My only concern is that this study only compared tree ring isotope measurements with satellite-based estimates of NPP but not in-situ NPP data, for instance, from forest inventories which can serve as ground-truth for validation purpose. To gain the readers the confidence of the use of MOD17 / GIMMS3g NPP product in this study, the author may refer to the performance of MOD17 NPP product over temperate forests (e.g., if there is any previous validation effort on space-based NPP estimates).

Thank you for the positive and constructive comments.

This is an important point that was also mentioned by Reviewer 1. The GIMMS3g NPP and MODIS NPP data were validated during their development against flux tower data over deciduous broadleaf forests (Smith *et al.*, 2016, Zhao & Running, 2010). In the revised version, we have rerun the analysis with the newest version of high-resolution terrestrial NPP products (MODIS NPP and Landsat NPP), which became available in March 2018 for the conterminous United States (Robinson *et al.*, 2018). The new MODIS and Landsat NPP used improved meteorological parameters and have been calibrated with empirical data from flux towers across the deciduous

broadleaf forest (Supplementary Fig. 4, see above). The performance of the new NPP products over deciduous broadleaf forest has improved considerably and both datasets match extremely well flux tower data.

Specific comment

L71: it is correct that we need long-term NPP proxy. Besides the method the authors developed in this study, I am just curious to know are there any other proxies (besides tree-ring width) that have been developed and tested? If so, the authors should review them and discuss the differences with the proxy they developed;

So far, NPP estimates come from three sources: 1) forest inventories, 2) flux towers, and 3) satellites. Space-based NPP are calculated by algorithms that are based on land cover, canopy reflectance and climatic data and are calibrated with land-based data (i.e. flux tower data). Forest inventories data provide information on biomass increment/changes in time but at very coarse temporal resolution.

In the text, we have reviewed these methods and highlighted their differences, advantages and disadvantages in the first paragraph and also at some places in the discussion section (lines 60-65, 233-239).

L127: the weaker correlation between tree-ring width and satellite NPP may indicate a lag response of wood productivity and annual NPP? not sure if would be worth to test lagged correlation between NPP and tree-width time series?

This is correct. It is possible that wood production lagged NPP and we also mentioned this in the text (lines 218-221). Mismatch between tree-ring width and ecosystem productivity was recently reported for a mixed conifer forest in northeastern US (Teets *et al.*, 2018). We tested lagged correlation between NPP and tree-ring width series and found weaker correlation values.

L342: the author may indicate which version of MOD17 product they used in this study (C5 or C6).

We have updated the analysis with the newest high-resolution MODIS NPP dataset available since March 2018 for the conterminous United States (Robinson *et al.*, 2018). It is the MOD17 version 6 data, which was refined to 250 m spatial resolution for the conterminous United States.

Fig. 1: the value appears to be too high for the slope of trend in T or P. What is the unit of trend? per year or per decade?

The trends in temperature and precipitation are for the whole period (1950-2015). This information is provided in the figure legend.

Fig. 3: maybe the author can also show time series of MOD17 NPP?

We have added the time series of Landsat NPP in the new figure 3. MODIS NPP time series are short (15 years) and correlations with tree-ring time series are weaker so we have decided to not include MODIS NPP in the new figure 3 for clarity.

References

- Brienen RJW, Gloor E, Clerici S *et al.* (2017) Tree height strongly affects estimates of water-use efficiency responses to climate and CO₂ using isotopes. *Nature Communications*, **8**, 288.
- Mascioli NR, Previdi M, Fiore AM, Ting M (2017) Timing and seasonality of the United States ‘warming hole’. *Environmental Research Letters*, **12**, 034008.
- Maxwell RS, Hessl AE, Cook ER, Pederson N (2011) A multispecies tree ring reconstruction of Potomac River streamflow (950–2001). *Water Resources Research*, **47**, 1-12.
- Pederson N, Bell AR, Cook ER *et al.* (2013) Is an epic pluvial masking the water insecurity of the greater New York City region? *Journal of Climate*, **26**, 1339–1354.
- Robinson NP, Allred BW, Smith WK *et al.* (2018) Terrestrial primary production for the conterminous United States derived from Landsat 30 m and MODIS 250 m. *Remote Sensing in Ecology and Conservation*, **doi: 10.1002/rse2.74**, 1-17.
- Smith WK, Reed SC, Cleveland CC *et al.* (2016) Large divergence of satellite and Earth system model estimates of global terrestrial CO₂ fertilization. *Nature Climate Change*, **6**, 306-310.
- Teets A, Fraver S, Hollinger DY, Weiskittel AR, Seymour RS, Richardson AD (2018) Linking annual tree growth with eddy-flux measures of net ecosystem productivity across twenty years of observation in a mixed conifer forest. *Agricultural and Forest Meteorology*, **249**, 479-487.
- Zhao M, Running SW (2010) Drought-induced reduction in global terrestrial net primary production from 2000 through 2009. *Science*, **329**, 940-943.

Reviewers' comments:

Reviewer #1 (Remarks to the Author):

Review of "Tree-ring isotopes reflect satellite-based net primary productivity estimates at the biome scale"

In response to an earlier submission of this manuscript, I was listed as Reviewer #1. As I said then, it's an interesting idea to use tree-ring isotopes to inform satellite-based models of NPP. I was trained as an ecophysiologicalist and I usually work with physiological mechanisms explaining ecological phenomena. Sometimes, we work with "hybrid" models, which combine physiological data with empirical factors adjusting model outputs. This manuscript takes that hybrid strategy even further, using physiological variables (tree-ring isotopes) to modify descriptive models. This seems like a reasonable thing to do.

However, I remain concerned about their choice of the response variable, modeled NPP, and the justification for that choice:

First, to be clear, when they say NPP, do they mean aboveground NPP? I assume so, but it seems odd that I need to ask. I assume so because belowground NPP is so very difficult to measure and there are no standard methods for doing it. I know that some models do it anyway. If this is one of those models, then some text describing how the model deals with these issues must be presented.

Second, the information in Suppl. Fig. 4 is not compelling. It is not acceptable to present GPP data to argue for the quality of NPP estimates, and certainly not for aboveground NPP. The difference matters because GPP comes before respiratory losses and the ratio of aboveground NPP to total NPP is quite variable.

Third, the spread in the data looks pretty bad. What is the R² of this curve? And what are the points? There are not 14000 eddy flux stations in the world.

Fourth, the quality of these modeled estimates of NPP is central to this analysis. This description of the response variable should be in the main manuscript, not in the supplementary materials.

Also, the authors still maintain that they are observing "strong" correlations (line 275). I don't see these as strong and I don't see why, theoretically, they would be strong. I asked if the satellite NPP was for cornfields in the Midwest. I presume that it is, which leads one to wonder how strong the correlations even should be considering that we're comparing a C4 grass to C3 trees. In their response to my question about this, they say that with data from more species, more spatial coverage, and longer time series, their estimates will improve. Given the problems in the response variable here, I'm skeptical.

It may be worth noting that the strong emphasis on VPD here may cause difficulty with any future isotope reviewers. In contrast to what is stated on line 194, Roden et al. do not discuss VPD. The Craig-Gordon model, which is the basis for their O-18 model, uses relative humidity as a variable. Although RH is correlated with VPD, the correlation varies seasonal and spatially. Perhaps this is a quibble, but it is an easy one to fix.

Reviewer #3 (Remarks to the Author):

The authors have nicely addressed all my comments on previous version. Since I am now very satisfied with the much improved quality of this submission and I am also convinced of the importance of findings reported, hence I would like to highly recommend it to be published in Nature Communications.

Reviewer #1 (Remarks to the Author):

Review of “Tree-ring isotopes reflect satellite-based net primary productivity estimates at the biome scale”

In response to an earlier submission of this manuscript, I was listed as Reviewer #1. As I said then, it's an interesting idea to use tree-ring isotopes to inform satellite-based models of NPP. I was trained as an ecophysiologicalist and I usually work with physiological mechanisms explaining ecological phenomena. Sometimes, we work with “hybrid” models, which combine physiological data with empirical factors adjusting model outputs. This manuscript takes that hybrid strategy even further, using physiological variables (tree-ring isotopes) to modify descriptive models. This seems like a reasonable thing to do.

Thank you very much for reviewing our manuscript and for your constructive comments. You will find below the responses to your comments with the description of the changes that we have made in the manuscript.

However, I remain concerned about their choice of the response variable, modeled NPP, and the justification for that choice:

First, to be clear, when they say NPP, do they mean aboveground NPP? I assume so, but it seems odd that I need to ask. I assume so because belowground NPP is so very difficult to measure and there are no standard methods for doing it. I know that some models do it anyway. If this is one of those models, then some text describing how the model deals with these issues must be presented.

You raised a very important point here and we apologize for the confusion. Process based models like the MOD17 and the underlying productivity products have been widely used in environmental sciences for estimating and monitoring the spatiotemporal variability in terrestrial ecosystem productivity (Running *et al.*, 2004; Zhao and Running, 2010; Cleveland *et al.*, 2015; Smith *et al.*, 2016). The satellite NPP product captures total NPP based on light use efficiency logic, and uses simple ratios of aboveground to belowground partitioning at the biome scale. The difficulty to measure belowground NPP and the paucity of aboveground NPP data make the calibration of satellite-based NPP data with field-based NPP measurements challenging. Still, there is evidence that roughly equal allocation of NPP between canopy, wood, and fine root growth exists, and these components are linearly related to total NPP (Aber and Federer, 1992; Gower *et al.*, 2001; Malhi *et al.*, 2011; Cleveland *et al.*, 2015). Additionally, there is evidence that NPP is a fixed proportion of GPP (Waring *et al.*, 1998), which supports the direct comparison between satellite-based NPP and eddy covariance flux tower-based GPP estimates. We acknowledge that there is a pressing need for dedicated field campaigns aimed at accurately estimating seasonal NPP and GPP dynamics, but for this work, we rely on the best available datasets. Of course, it would be more accurate to compare satellite aboveground NPP to flux tower aboveground NPP to isotope aboveground NPP, but this is not possible and a limitation of the current suite of productivity measurement techniques that we have developed as a field. However, despite this limitation, it is quite remarkable that these independent data correspond well.

In order to address this concern, we have provided more detailed information about the characteristics of the satellite NPP data and how these data have been calibrated with ground measurement of NPP. Specifically, we have added two new text sections in the Methods describing the NPP algorithm used for producing the satellite-based NPP estimates and the limitations that exist in the field as a whole (lines 386-443).

Second, the information in Suppl. Fig. 4 is not compelling. It is not acceptable to present GPP data to argue for the quality of NPP estimates, and certainly not for aboveground NPP. The difference matters because GPP comes before respiratory losses and the ratio of aboveground NPP to total NPP is quite variable.

Using flux tower GPP to validate satellite GPP in place of validating satellite total NPP is a very common method of validation, and the only one that currently exists because field estimates of NPP are not widely available. Even when they are available, there is typically a lot of bias/error associated with field collections of NPP data (Cleveland *et al.*, 2015). There are many studies that use flux tower GPP to validate satellite GPP instead of satellite NPP (e.g. Running *et al.*, 2004; Zhao *et al.*, 2005; Jung *et al.*, 2011; Jones *et al.*, 2017; Robinson *et al.*, 2018). Flux tower GPP is treated as the standard procedure since it should be low bias/error and because typically the GPP:NPP ratio is often assumed to stay relatively fixed (Waring *et al.*, 1998). Although this assumption is an interesting area of research, it is outside the scope of our research in this manuscript. To our knowledge, we did the best approach possible at the moment by using the latest generation of satellite-based productivity estimates that were intensively validated with 43 eddy covariance flux towers. These datasets were peer-reviewed and have been publicly available since March 2018.

Third, the spread in the data looks pretty bad. What is the R² of this curve? And what are the points? There are not 14000 eddy flux stations in the world.

We apologize for the confusion regarding the supplementary Figure 5. Each point represents an 8-day mean GPP estimate from eddy covariance flux towers located in deciduous broadleaf forests in eastern US versus modeled 8-day GPP value calculated using the MODIS MOD17 algorithm at these flux tower sites. A total 14 533 measurements were used for the validation of the MODIS and Landsat GPP datasets over deciduous broadleaf forests. The Modis and Landsat GPP datasets account for 90 and 88% of the variance from the flux tower measurements, respectively. We have added all this information in the figure legend and the R² values (MODIS, R²=0.903; Landsat, R²=0.884) in inset (new Supplementary Fig. 5).

Fourth, the quality of these modeled estimates of NPP is central to this analysis. This description of the response variable should be in the main manuscript, not in the supplementary materials.

We totally agree with the reviewer. This is a very important point and we apologize of not having including information on the calculation of satellite NPP in the main manuscript. In the revised version, we have added a paragraph explaining the MODIS MOD17 algorithm used for the calculation of the satellite NPP estimates and discussed its limitations (Section MOD17 algorithm and limitations, lines 386-443).

Also, the authors still maintain that they are observing “strong” correlations (line 275). I don’t see these as strong and I don’t see why, theoretically, they would be strong. I asked if the satellite NPP was for cornfields in the Midwest. I presume that it is, which leads one to wonder how strong the correlations even should be considering that we’re comparing a C4 grass to C3 trees. In their response to my question about this, they say that with data from more species, more spatial coverage, and longer time series, their estimates will improve. Given the problems in the response variable here, I’m skeptical.

This is a good point. We would first like to note that the strongest correlations occur mostly over forested regions (e.g. northeastern US, southern Appalachian). Specifically, the spatial field correlation analysis indicates high agreements between satellite NPP time series and tree-ring isotope chronologies as indicated by field correlations values ≤ 0.6 ($\delta^{18}\text{O}$, Fig. 2a) and ≥ 0.6 ($\Delta^{13}\text{C}$, Fig. 2b) above the forested northeastern US and Appalachian. At the local scale, we report also high agreements between Landsat NPP and tree-ring $\delta^{18}\text{O}$ at Frick Creek ($r = -0.71$), and tree-ring $\Delta^{13}\text{C}$ at Black Rock Forest ($r = 0.72$). Such correlations can be considered as strong.

That said, the NPP datasets from the MOD17 algorithm uses 30-m resolution land cover type from the National Land Cover Database as input data. Depending on the land cover classification, each 30-m pixel can represent different vegetation types (e.g. cropland, deciduous broadleaf forest). For the spatial field correlation analysis, we did not specifically distinguish between land cover/vegetation types as we wanted to assess if tree-ring time series (width, isotopes) correlate with pixel wise satellite NPP across the eastern US. In other words, this analysis was used to test the similarity in interannual variability between remotely sensed NPP and tree-ring width and isotope chronologies irrespective to vegetation types. This analysis does not inform on absolute changes in NPP since NPP absolute values are different between vegetation types (e.g. C4 cornfield vs C3 broadleaf forest). The take home message of our study is that tree-ring isotopes capture interannual vegetation productivity dynamics at the biome scale irrespective of vegetation types.

It may be worth noting that the strong emphasis on VPD here may cause difficulty with any future isotope reviewers. In contrast to what is stated on line 194, Roden et al. do not discuss VPD. The Craig-Gordon model, which is the basis for their O-18 model, uses relative humidity as a variable. Although RH is correlated with VPD, the correlation varies seasonal and spatially. Perhaps this is a quibble, but it is an easy one to fix.

Thank you for pointing this out. We cited the wrong reference here. We have changed Roden et al (2000) for Kahmen et al (2011). The work of Kahmen et al (2011) clearly showed that the individual impacts of relative humidity and air temperature on $\delta^{18}\text{O}$ values of cellulose is best explained when integrating both environmental factors into a single index estimating the atmospheric vapor demand experienced by plants (VPD) (lines 198-202).

Reviewer #3 (Remarks to the Author):

The authors have nicely addressed all my comments on previous version. Since I am now very satisfied with the much improved quality of this submission and I am also convinced of the importance of findings reported, hence I would like to highly recommend it to be published in Nature Communications.

Thank you for the constructive review and positive feedback. Your comments have been very valuable for the improvement of our study.

References

- Aber, J.D., Federer, C.A., 1992. A generalized, lumped-parameter model of photosynthesis, evapotranspiration and net primary production in temperate and boreal forest ecosystems. *Oecologia* 92, 463-474.
- Cleveland, C.C., Taylor, P., Chadwick, K.D., Dahlin, K., Doughty, C.E., Malhi, Y., Smith, W.K., Sullivan, B.W., Wieder, W.R., Townsend, A.R., 2015. A comparison of plot-based satellite and Earth system model estimates of tropical forest net primary production. *Global Biogeochemical Cycles* 29, 626-644.
- Gower, S.T., Krankina, O., Olson, R.J., Apps, M., Linder, S., Wang, C., 2001. Net primary production and carbon allocation patterns of boreal forest ecosystems. *Ecological Applications* 11, 1395-1411.
- Jones, L.A., Kimball, J.S., Madani, N., Reichle, R., Glassy, J., Ardizzone, J., 2017. The SMAP level 4 carbon product for monitoring terrestrial ecosystem-atmosphere CO₂ exchange. *IEEE Transactions on geoscience and remote sensing* DOI: 10.1109/TGRS.2017.2729343.
- Jung, M., Reichstein, M., Margolis, H.A., Cescatti, A., Richardson, A.D., Arain, M.A., Arneth, A., Bernhofer, C., Bonal, D., Chen, J., Gianelle, D., Gobron, N., Kiely, G., Kutsch, W., Lasslop, G., Law, B.E., Lindroth, A., Merbold, L., Montagnani, L., Moors, E.J., Papale, D., Sottocornola, M., Vaccari, F., Williams, C., 2011. Global patterns of land-atmosphere fluxes of carbon dioxide, latent heat, and sensible heat derived from eddy covariance, satellite, and meteorological observations. *Journal of Geophysical Research: Biogeosciences* 116, doi:10.1029/2010JG001566.
- Kahmen, A., Sachse, D., Arndt, S.K., Tu, K.P., Farrington, H., Vitousek, P.M., Dawson, T.E., 2011. Cellulose $\delta^{18}\text{O}$ is an index of leaf-to-air vapor pressure difference (VPD) in tropical plants. *Proc. Natl Acad. Sci. USA* 108, 1981–1986.
- Malhi, Y., Doughty, C., Galbraith, D., 2011. The allocation of ecosystem net primary productivity in tropical forests. *Philosophical Transactions of the Royal Society B: Biological Sciences* 366, 3225-3245.
- Robinson, N.P., Allred, B.W., Smith, W.K., Jones, M.O., Moreno, A., Erickson, T.A., Naugle, D.E., Running, S.W., 2018. Terrestrial primary production for the conterminous United States derived from Landsat 30 m and MODIS 250 m. *Remote Sensing in Ecology and Conservation*, doi:10.1002/rse1002.1074.
- Roden, J.S., Lin, G., Ehleringer, J.R., 2000. A mechanistic model for interpretation of hydrogen and oxygen isotope ratios in tree-ring cellulose. *Geochimica et Cosmochimica Acta* 64, 21–35.
- Running, S.W., Nemani, R.R., Heinsch, F.A., Zhao, M., Reeves, M., Hashimoto, H., 2004. A continuous satellite-derived measure of global terrestrial primary production. *BioScience* 54, 547-560.
- Smith, W.K., Reed, S.C., Cleveland, C.C., Ballantyne, A.P., Anderegg, W.R.L., Wieder, W.R., Liu, Y.Y., Running, S.W., 2016. Large divergence of satellite and Earth system model estimates of global terrestrial CO₂ fertilization. *Nature Clim. Change* 6, 306-310.

Waring, R.H., Landsberg, J.J., Williams, M., 1998. Net primary production of forests: a constant fraction of gross primary production? *Tree Physiol* 18, 129-134.

Zhao, M., Heinsch, F.A., Nemani, R.R., Running, S.W., 2005. Improvements of the MODIS terrestrial gross and net primary production global data set. *Remote Sensing of Environment* 95, 164-176.

Zhao, M., Running, S.W., 2010. Drought-induced reduction in global terrestrial net primary production from 2000 through 2009. *Science* 329, 940-943.

REVIEWERS' COMMENTS:

Reviewer #1 (Remarks to the Author):

The description of the methods and the discussion are much improved and I now have a clearer understanding of what was done and why. But I still am not convinced that untested, modeled estimates of NPP from remote sensing provide a reliable test of hypotheses. The assumptions in these models are far less certain than these authors recognize.

The first section of the instructions for reviewers asks if the manuscript "provides strong evidence for its conclusions." It does not, and I don't see how the authors can change that. I understand that the remote sensing and modeling communities have greater faith in the generality and reliability of their tools, so maybe the best way forward is to send this manuscript there.

Reviewer #1 (Remarks to the Author):

The description of the methods and the discussion are much improved and I now have a clearer understanding of what was done and why. But I still am not convinced that untested, modeled estimates of NPP from remote sensing provide a reliable test of hypotheses. The assumptions in these models are far less certain than these authors recognize.

The first section of the instructions for reviewers asks if the manuscript "provides strong evidence for its conclusions." It does not, and I don't see how the authors can change that. I understand that the remote sensing and modeling communities have greater faith in the generality and reliability of their tools, so maybe the best way forward is to send this manuscript there.

Thank you for your review and critical feedback.

We are well aware of the limitations of the assumptions made in the modelled NPP estimates from remote sensing and we now acknowledge the limitations of the assumptions in the discussion section of the manuscript (lines 288-320). We would also like to emphasise that the modelled satellite-based NPP data used in our study have been calibrated and tested with measured ecosystem productivity data from flux towers located across the eastern US broadleaf deciduous forest – our study region (supplementary Fig. 5). We have used for our analyses the most recent, state-of-the-art NPP datasets that have been calibrated and independently tested. These datasets have been peer-reviewed and used in many other environmental and ecological studies. We therefore rather disagree with the Reviewer's comment that questions the reliability of the NPP products used in our study and recent improvements in calculation of remotely sensed NPP estimates. Besides clearly mentioning the limitations of the assumptions for the calculation of the remotely-sensed NPP data, we also want to emphasise that our tree-ring data can be seen as an independent proxy for NPP that could be further used for the calibration of remotely-sensed NPP.